# Remote Teaching, Self-Resilience, Stress, Professional Efficacy, and Subjective Health among Israeli PE Teachers during the COVID-19 Pandemic



**Ronit Ben Amotz** [1,2] , **Gizell Green** [3], **Gili Joseph** [4], **Sharon Levi** [5], **Niva Manor** [5], **Kwok Ng** [6,7], **Sharon Barak** [3], **Yeshayahu Hutzler** [8] and **Riki Tesler** [1,*]

1  Department of Health Systems Management, School of Health Sciences, Ariel University, Ariel 4077625, Israel; ronit@benamotz.com
2  Health Promotion & Wellbeing Research Center, Ariel University, Ariel 4077625, Israel
3  Department of Nursing, School of Health Sciences, Ariel University, Ariel 4077625, Israel; gizellgr@ariel.ac.il (G.G.); sharoni.baraki@gmail.com (S.B.)
4  Department of Physical Education, Faculty of Health & Science, Kibbutzim College of Education, Technology and the Arts, Tel Aviv 6250769, Israel; gili.joseph@smkb.ac.il
5  Department of Health Systems Management, School of Public Health, Haifa University, Haifa 2611001, Israel; sharonkahanelevi@gmail.com (S.L.); nivamanor@gmail.com (N.M.)
6  Physical Activity for Health Research Cluster, Department of Physical Education and Sport Sciences, University of Limerick, V94 T9PX Limerick, Ireland; kwok.ng@jhbsc.org
7  School of Educational Sciences and Psychology, University of Eastern Finland, 80100 Joensuu, Finland
8  Department of Physical Education, Wingate College, Netanya 42902, Israel; shayke.hutzler@gmail.com
*  Correspondence: riki.tesler@gmail.com

**Abstract:** This study investigated demographic factors, teaching characteristics, psychological characteristics, school-related characteristics, professional efficacy, and subjective health perceptions among PE teachers during the COVID-19 pandemic. We conducted a cross-sectional research design. Questionnaires were distributed to PE teachers online during COVID-19 closures. PE teachers (N = 757) from elementary, middle, and high schools in Israel voluntary completed surveys on the topics of stress levels, self-resilience, remote teaching, professional efficacy, and subjective health perception. Sex, remote-teaching experience and clear remote school policy significantly predicted professional efficacy. Sex, teaching experience and self-resilience significantly predicted subjective health perception. This study demonstrated the need for a clear remote policy, as it likely empowers teacher professional efficacy. Transparent procedures and guidelines, along with clarifying remote policies by a supportive administration, are important for the professional efficacy of PE teachers. In addition, educational programs that are aimed at developing and strengthening the values of a healthy, positive, and balanced lifestyle are important to subjective health perception among PE teachers.

**Keywords:** remote teaching; professional efficacy; subjective health; PE teachers

## 1. Introduction

In March 2020, the World Health Organization announced the SARS-CoV-2 outbreak, the virus causing COVID-19 disease, as a global pandemic [1]. Restrictions were quickly put into place worldwide, causing a significant disruption to daily life, including within education systems, affecting students and teachers alike [2]. Such restrictions forced many institutions to cease in-person teaching at short notice [3], resulting in remote teaching [4].

The transition to remote teaching was not typical and could be described as emergency remote teaching, defined as "a temporary shift of instructional delivery to an alternate delivery model due to crisis circumstances" [5]. Contrary to online learning, emergency remote teaching is a temporary solution that serves as an alternative to face-to-face teaching and ceases as soon as the state of emergency subsides. However, in light of the pandemic,

teachers' unusual circumstances prevented them and their students from making a more natural transition to remote education [6] and some concerns have been raised in regard to their ability to acquire the necessary competences required for effective instruction in a digital environment [7]. Remote-teaching methods allow the teacher to instruct theoretical study material without the social values that are normally integrated into the curriculum [8]. Furthermore, empirical evidence suggests specific difficulties in teaching technical skills such as laboratory skills [8,9]. Likewise, among the different subjects studied in school, physical activity (PE) is one of the most challenging to teach remotely, as lessons are based on hands-on activities that promote social values. Although the challenges of teaching PE remotely have been discussed through the use of different technologies, less is known about the emergency circumstances and their effect on PE teachers.

*Professional efficacy* among educators has been described as the level of confidence a teacher possesses regarding his or her ability to enable pupils to progress toward their desired outcomes. Research has shown that teachers' sense of efficacy is related to pupils' achievement, motivation, and efficacy [9]. For most teachers, teaching experience increases effectiveness. As teachers gain experience, their students are more likely to perform better on measures of success beyond test scores, such as school attendance [10].

As opposed to teachers with low professional efficacy, those with high professional efficacy tend to devote greater effort towards their goals, planning, organization, teaching, and aspirations, and are often more open-minded to new pedagogy and stay in the profession longer. When staff members believe that they can stimulate progress and challenge their pupils together, it has the biggest impact on teacher's achievement and determination when facing a challenge [11]. Teachers with low professional efficacy tend to experience higher stress levels associated with their profession [12], leading to dropping out of the profession [13].

Professional efficacy can affect mental health, which plays a role in individuals' thinking modes, decision making, quality of encountering problems, depression levels, anxiety status, and more [14]. Health is a state of complete physical, mental, and social wellbeing, not merely the absence of disease [15]. Subjective health perception is explicitly coupled with wellbeing and considers physical, psychological, and social factors. During the beginning of the COVID-19 pandemic, educators faced unprecedented challenges, including the disruption of established instructional programs and routines with the rapid transition from in-person teaching to remote learning, along with the emotional toll of isolation due to social-distancing efforts, and uncertainty about personal safety and health [16].

During the pandemic, planning and implementing PE lessons fell on the PE teachers themselves [17]. The experience of teachers who taught PE remotely varied by the types of schools in which they taught, regions where they were located, and school grade [17]. A recent study has shown that the majority (92.4%) of teachers indicated that they had never taught online before the emergency transition, and very few had received any meaningful training from their school or school district [18]. The use of technology in teaching PE remotely, such as movement analysis, was starting to slowly emerge before the pandemic, but it is possible that only new teachers were aware of such methods [19]. Research conducted before the outbreak of COVID-19 suggests that many PE teachers felt unprepared to use technology in lessons [20].

As these new methods were unknown to the majority of teachers, it is possible they caused stress, which often occurs when teachers feel that the job's demands become too much to manage [21]. To deal with such stress, it is important for teachers to have self-resilience, which is instrumental in navigating personal growth and satisfaction. These skills are developed over a lifetime when dealing with and overcoming various challenges [22].

PE teachers were left with unclear policies and support regarding the implementation of effective remote teaching methods [17]. Remote PE classes tend to offer limited content, with an emphasis on fitness, health, and weight training. In general, PE encourages an increase in students' engagement in physical activity. However, one (pre-COVID-19) study has shown that most PE classes taught remotely did not meet curriculum guidelines and

even reduced students' physical activity [23]. In many cases, teachers used trial-and-error methods in implementing remote instruction during the COVID-19 pandemic [24,25].

The aim of this study was to examine the association between demographic factors, teaching characteristics, psychological characteristics, school-related characteristics, and subjective health perception among Israeli PE teachers during the COVID-19 pandemic.

## 2. Materials and Methods

### 2.1. Research Design

This was a cross-sectional study design. Questionnaires were distributed to PE teachers online during COVID-19 closures. Questionnaires examined the exact variables, including professional efficacy and teachers' resilience, considering the last two closures.

### 2.2. Participants

We conducted a convenience research sample: from a list of PE teachers provided by the Israeli Ministry of Education, we asked teachers to fill out the questionnaire. The selection included 800 teachers from regular- and special-education schools. Inclusion criteria were PE teachers who had taught PE for at least one year in schools and agreed to participate in the study. The exclusion criteria were PE teachers who taught less than one year and those who taught in other learning areas. The final study population included 757 PE teachers [26].

### 2.3. Data Collection

A letter was sent to the relevant teachers explaining the learning objectives and their contribution to this study. Selected teachers filled out online questionnaires between May and June 2020, and responses were collected anonymously by the lead researcher. This study was approved by the Ethics Committee of Israel's Chief Scientist Office and the Israeli Ministry of Education.

### 2.4. Independent Variables

Socio-demographic variables included sex (female, male), age, family status (single, in a relationship, divorced, widowed), religion (Jewish, Muslim, Christian, Druze, other), level of religiosity (secular, traditional, religious; measured by self-definition), education level (senior teacher, Bachelor's degree, Master's degree, Doctoral degree), and years of seniority at school.

Teaching variables included school state, religion (Jewish, orthodox, Arab, Druze) and school type (elementary, middle, high school).

Policy regarding remote teaching was measured by items on the EUFAPA (European Federation of Adapted Physical Activity) survey. One item included "In my workplace, there are clear guidelines regarding the usage of remote teaching and remote communication." Answers ranged from 1 = completely disagree to 5 = strongly agree.

Remote-teaching media usage was measured by items on the EUFAPA survey. One question included "What media do you use daily to communicate with students and colleagues?" Another was "What media do you use for remote teaching in PE lessons?" [27].

### 2.5. Dependent Variables

The *teacher's stress* measure, the dependent variable, was assessed by an index with 12 items [28]. Items from the index were measured using a 5-degree Likert scale, with 1 = weak feeling of stress and 5 = a strong sense of stress, while 3 = indifference. For example: "Typically, in your opinion, how stressful is your work?".

*Subjective wellbeing* was measured by a questionnaire that was initially developed by Veit and Ware (1983) [29] and validated by Chen, Gully and Eden (2001) [30]. The questionnaire has ten questions and answers are measured on a Likert scale of 1–5, where 1 = strongly disagree and 5 = strongly agree. For example: "Overall, how do you define your health condition?".

*Professional efficacy* (in the context of remote teaching) was measured by a questionnaire, with answers measured on a Likert scale where 1 = strongly disagree and 5 = strongly agree. For example: "It is vital that I succeed in delivering content in PE via remote teaching." The questionnaire has been tested in various studies and has a high content validity and predictive validity [30].

*Resilience* was measured using a Brief Resilience Scale that includes six items on a 5-point Likert scale, ranging from 1 = strongly disagree to 5 = strongly agree. For example: "I have a hard time overcoming stressful events"; "It is difficult for me to overcome when something terrible happens" [31].

### 2.6. Data Analysis

Descriptive statistics (means, standard deviations, and percentages) were used to describe participants' demographic and teaching characteristics. Chi-squared tests of independence were used to examine differences in the prevalence of demographic and teaching characteristics by sex group.

*Factors associated with professional efficacy and subjective health perception:*

Since professional efficacy and subjective health perception were not normally distributed (Shapiro–Wilk test for normal distribution = 0.984 and 0.90, respectively; $p < 0.001$), non-parametric statistics were used to evaluate related factors. More specifically, correlations between the variables and demographic, teaching, psychological, and school-related characteristics were examined using the Spearman rank correlation coefficient. Differences between males and females were examined using the Mann–Whitney test.

*Factors predicting professional efficacy and subjective health perception:*

Enter multiple linear regression analysis was used for evaluating factors predicting professional efficacy. Binary logistic regression modeling was used to determine the extent to which the independent variables were predictive of subjective health perception. In that respect, the dependent variables (i.e., subjective health perception) were coded as 0, not reporting excellent health (health described as: "not that good", "good", or "very good") and 1, reporting excellent health [32]. In both regression models, only variables with significant correlations with the dependent variables were included. All independent variables were checked for multi-collinearity using the variance of inflation factor >10 [33]. The criterion for inclusion in the model was an $\alpha$ level of 0.05, and the exclusion criterion was an $\alpha$ level of 0.10.

Post-hoc power analysis for the regression analysis (test family—F tests; statistical test—linear multiple regression: fixed models, $R^2$ increase; type of power analysis—compute achieved power given $\alpha$ sample size, and effect size) was conducted. The regression analysis included only variables that significantly correlated with professional efficacy and subjective health perception. For professional efficacy, seven predictors were included in the regression analysis. Based on the mean correlations of the predictors with professional efficacy, partial $R^2$ was 0.03 (i.e., small effect size). Based on these statistical values and a sample size of 757, the power to predict professional efficacy was 0.90. For subjective health perception, five predictors were included in the analysis. Based on the mean correlations of the predictors with professional efficacy, partial $R^2$ was 0.02 (i.e., small effect size). Based on these statistical values and a sample size of 757, the power to predict subjective health perception was 0.86. Data analyses were conducted using SPSS version 25 (SPSS Inc., Chicago, IL, USA). In all analyses, the significance level was set at 0.05 (two-tailed).

### 3. Results

A total of 757 PE teachers participated in this study (mean age = 44.08 + 10.26 years; 65.90% female). Participants' mean teaching experience was 17.45 + 11.06 years. In demographic characteristics, compared to males, the proportion of "single" and "Jewish" in females was statistically significantly greater (Chi-square = 10.00 and 58.75, respectively; $p < 0.01$). Similarly, in teaching characteristics, when compared to males, the proportion of females who taught in "state schools" and in "primary and elementary schools"

was statistically significantly greater (Chi-square = 28.40 and 7.11, respectively; $p < 0.001$; Table 1).

**Table 1.** Chi-square tests of independence for demographic and teaching characteristics, by sex (N = 757).

| Variables | | Female | Male | Chi-Square (*p*-Value) |
|---|---|---|---|---|
| | | N = 499 % | N = 258 % | |
| **Demographics** | Family status | | | |
| | Single | 12.8 | 7.8 | 10.00 (0.01) |
| | Married | 78.0 | 86.8 | |
| | Divorced | 8.2 | 5.4 | |
| | Widowed | 1.0 | 0.0 | |
| | Religion | | | |
| | Jewish | 91.2 | 72.9 | 58.75 (<0.001) |
| | Christian | 2.2 | 2.3 | |
| | Muslim | 3.8 | 20.2 | |
| | Druze | 1.4 | 3.5 | |
| | Other | 1.4 | 1.2 | |
| | Religiosity | | | |
| | Secular | 64.3 | 62.9 | 0.36 (0.83) |
| | Traditional | 21.7 | 23.6 | |
| | Religious | 14.0 | 13.4 | |
| | Education level | | | |
| | Senior teacher | 3.2 | 1.9 | 2.86 (0.41) |
| | Bachelor's degree | 52.7 | 54.7 | |
| | Master's degree | 43.5 | 41.9 | |
| | Doctoral degree | 0.6 | 1.6 | |
| **Teaching characteristics** | School type | | | |
| | State school | 76.2 | 68.2 | 28.40 (<0.001) |
| | Religious school | 15.4 | 11.2 | |
| | Orthodox school | 0.8 | 0.8 | |
| | Arab school | 5.8 | 17.8 | |
| | Druze school | 1.8 | 1.9 | |
| | School type | | | |
| | Primary and elementary school | 46.7 | 37.2 | 7.11 (0.007) |
| | Middle school | 27.2 | 33.1 | |
| | High school | 26.1 | 29.7 | |

Notes: For school grade, the total number exceeds the number of females or males, as several teachers teach more than one school grade.

**Professional efficacy and subjective health perception levels**

Participants' professional efficacy varied greatly and ranged from 1.36 to 4.90 points with a median score of 3.63. Lower to upper quartile (25 to 75 percentile) scores ranged from 3.18 to 4 points.

Statistically significant differences were observed in the prevalence of the various subjective health perception statuses (Chi-squared = 466.36; $p < 0.001$). More specifically, 48.2 (n = 365), 39.9 (n = 302), 11.1 (n = 84), and 0.8% (n = 6) of the sample presented "excellent", "very good", "good", and "not that good" subjective health, respectively. No study participant reported "not good at all" health.

Statistically significant associations were found between all the examined variables and professional efficacy (r range: 0.07 to 0.57; $p < 0.01$; Table 2).

**Table 2.** Factors associated with professional efficacy (Spearman rank correlation coefficients) (N = 757).

|  |  | Professional Efficacy |
| --- | --- | --- |
| Demographic characteristics | Age, years | −0.09 * |
| Teaching characteristics | Teaching experience, years | −0.07 * |
|  | Remote-teaching experience, average score | 0.57 * |
| Psychological characteristics | Self-resilience, average score | 0.10 * |
|  | Stress during the pandemic, average score | −0.16 * |
| School-related characteristics | Remote-teaching school policy, average score | 0.31 * |

Notes: * Significant correlation at *p* < 0.01 (Spearman rank correlation coefficient).

Females had higher professional efficacy than males (females: median score = 3.63, average rank = 403.59; males: median score = 3.50, average rank = 331.42; *p* < 0.001; Figure 1).

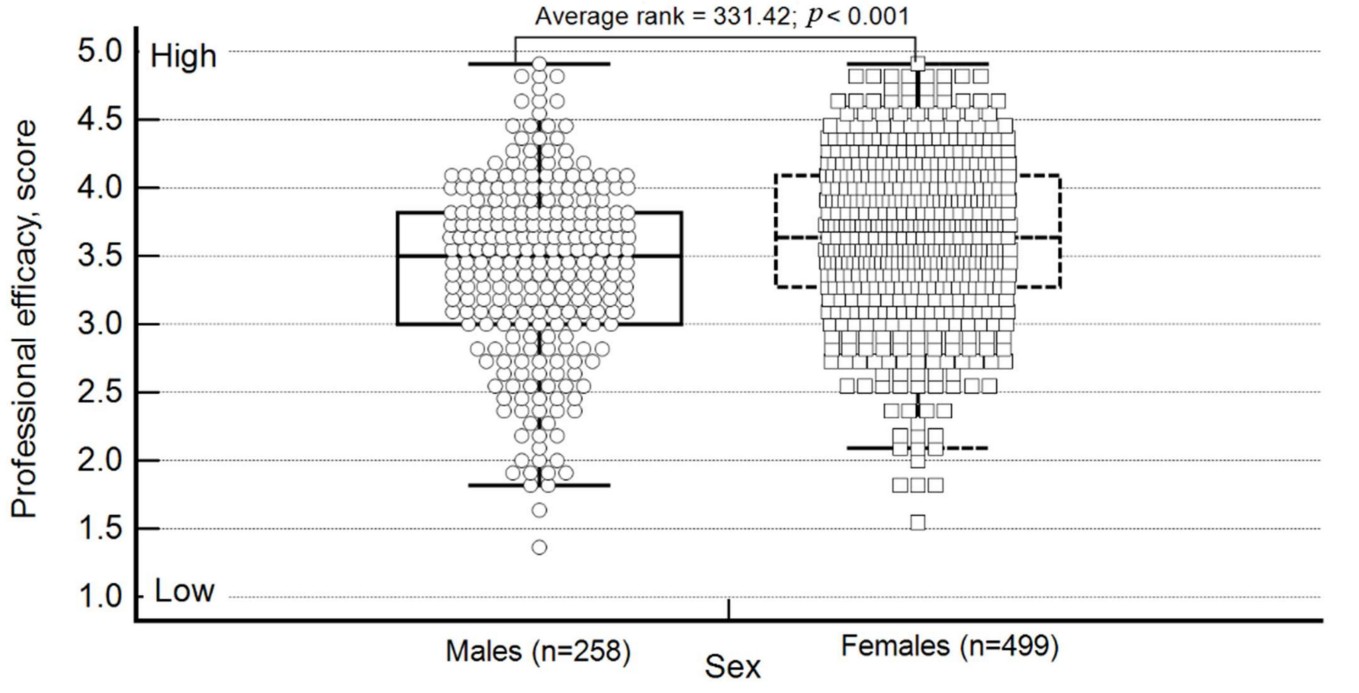

**Figure 1.** Professional efficacy: sex differences. Notes: The central box represents the values from the lower to upper quartile (25–75 percentiles). The vertical line extends from the minimum to the maximum value, excluding outside values, which are displayed as separate points. An outside value is defined as a value that is smaller than the lower quartile minus 1.5 times the interquartile range, or larger than the upper quartile plus 1.5 times the interquartile range. The middle line represents the median.

However, according to multiple logistic regression, only three variables significantly predicted professional efficacy: sex, remote-teaching experience, and remote school policy (F-ratio: 68.81; *p* < 0.001). More specifically, being female, having greater remote-teaching experience, and a clearer school policy all predicted higher professional efficacy. Overall, the model explained 39% of the variability of professional efficacy (adjusted R$^2$ = 0.39; Table 3).

Subjective health perception was significantly correlated (*p* < 0.01) with all variables, except for school policy (r range: 0.06 to 0.22; Table 4).

**Table 3.** Predictors of professional efficacy (multiple linear regression analysis).

|  | Independent Variable | Coefficient | Standard Error | t | *p* | Variance Inflation Factor |
|---|---|---|---|---|---|---|
|  | (Constant) | 1.70 |  |  |  |  |
| Demographic characteristics | Age, years | −0.00 | 0.00 | −1.64 | 0.09 | 5.32 |
|  | Sex (in comparison to males) | 0.16 | 0.03 | 4.08 | <0.001 | 1.04 |
| Teaching characteristics | Teaching experience, years | 0.00 | 0.00 | 0.99 | 0.32 | 5.33 |
|  | Remote-teaching experience, average score | 0.45 | 0.02 | 16.91 | <0.001 | 1.22 |
| Psychological characteristics | Self-resilience, score | 0.04 | 0.02 | 1.57 | 0.11 | 1.05 |
|  | Stress during the pandemic, average score | −0.00 | 0.00 | −0.42 | 0.66 | 1.11 |
| School-related characteristics | Remote school policy, average score | 0.07 | 0.01 | 3.83 | 0.0001 | 1.17 |
| Model summary | | F-ratio: 68.81 (7742); adjusted R$^2$ = 0.39; *p* < 0.0001 | | | | |

Notes: The inclusion criteria of the model were an F probability of 0.05 and exclusion criteria was an F probability of 0.1; only variables that had significant correlations with the dependent variable were included. Variables were entered in order of correlation's strength.

**Table 4.** Factors associated with subjective health perception (Spearman rank correlation coefficients; N = 757).

|  |  | Subjective Health Perception |
|---|---|---|
| Demographic characteristics | Age, years | −0.21 * |
| Teaching characteristics | Teaching experience, years | −0.22 * |
|  | Remote-teaching experience, average score | 0.08 * |
| Psychological characteristics | Self-resilience, average score | 0.16 * |
|  | Stress during the pandemic, average score | −0.09 * |
| School-related characteristics | Remote-teaching school policy, average score | 0.06 |

Notes: * Significant correlation at *p* < 0.01 (Spearman rank correlation coefficient).

The binary logistic regression model showed that being a female (odds ratio [OR] = 1.39; *p* = 0.04), greater teaching experience (OR = 0.96; *p* = 0.04), and higher self-resilience (OR = 1.64, *p* < 0.001) all statistically significantly predicted subjective health perception (chi-squared = 67.88, *p* < 0.001, Nagelkerke R$^2$ = 0.12; Table 5).

**Table 5.** Summary of multiple binary logistic regression analysis for prediction of subjective health perception.

| Predictor | Coefficient | Standard Error | Odds Ratio | Wald | 95% CI | *p*-Value |
|---|---|---|---|---|---|---|
| Constant | −0.98 | 0.80 |  | 1.20 |  | 0.04 |
| Sex (Reference, males) | 0.33 | 0.16 | 1.39 | 3.92 | 1.00–1.92 | 0.04 |
| Age, years | −0.01 | 0.01 | 0.98 | 0.86 | 0.95–1.01 | 0.35 |
| Teaching experience, years | −0.03 | 0.01 | 0.96 | 4.03 | 0.93–0.99 | 0.04 |
| Remote-teaching experience, average score | 0.06 | 0.10 | 1.06 | 0.33 | 0.86–1.31 | 0.56 |
| Self-resilience, average score | 0.49 | 0.11 | 1.64 | 20.00 | 1.32–2.04 | <0.001 |
| Stress during the pandemic, average score | −0.02 | 0.03 | 0.97 | 0.68 | 0.91–1.03 | 0.40 |
| Model summary | | Chi-squared = 67.88, *p* < 0.001, Nagelkerke R$^2$ = 0.12 | | | | |

Notes: The inclusion criteria of the model were an F probability of 0.05 and exclusion criteria was an F probability of 0.1; only variables that had significant correlations with the dependent variable were included; variables were entered in order of correlation's strength. Abbreviation: CI, confidence interval.

## 4. Discussion

We examined the associations and predictors of having a remote-teaching policy, remote-teaching experience, professional efficacy, self-resilience, stress, and subjective health among PE teachers during COVID-19. Our findings showed statistically significant associations between all the examined variables and professional efficacy. Moreover, we found our first hypothesis to be accurate, in that female teachers had higher professional efficacy than male teachers. Prior studies corroborate our results, indicating that female teachers had higher self-efficacy as compared to their male counterparts [34–36].

Research conducted before the COVID-19 pandemic has suggested that many PE teachers feel unprepared to use technology [37]. Hence, with the sudden outbreak of

the pandemic, PE teachers were left with unclear policies and support regarding the implementation of effective remote teaching [17].

Confirming our second and third hypotheses, our findings revealed that three variables significantly predicted professional efficacy: sex, remote-teaching experience, and remote school policy. More specifically, we found that being female, having greater remote-teaching experience, and a clearer school policy all predicted higher professional efficacy. Thus, the better understanding of teaching PE remotely is crucial, as it is unclear how long these practices will be in place. Hence, providing accessible online professional development programs that include technical skills and knowledge along with supportive school management and a transparent remote-teaching policy is highly recommended. Furthermore, a recent study found that clear policies and direct guidance need to be provided, as well as increased professional development and access to devices and adequate infrastructure, which may increase teacher professional efficacy, self-resilience, and wellbeing [38].

Consistent with the fourth hypothesis, that teachers with higher levels of resilience will report higher subjective health perceptions than teachers with low levels of resilience, our findings showed that subjective health perception was significantly correlated with all variables except for school policy. The characteristics that were correlated with subjective health (i.e., socio-demographic, teaching, and psychological characteristics) are all related to the self, whereas school-related characteristics were not correlated. This may imply that inner characteristics have more influence on subjective health perceptions than external characteristics. In particular, the binary logistic regression model showed that being a female, having more teaching experience, and higher self-resilience all statistically significantly predicted subjective health perception. These results are in line with a recent study indicating that a blended inquiry-based stress reduction intervention enhanced the resilience and improved the subjective and psychological wellbeing of teachers and showed positive correlations between resilience and psychological wellbeing [39]. Another study on Greek secondary teachers' resilience and occupational wellbeing showed that teachers' resilience correlated positively with their occupational wellbeing and that teachers' scientific specialization was related to their resilience levels [40].

Teacher resilience is conceived as being characterized by job satisfaction, commitment, teaching efficacy, motivation, wellbeing, and a professional sense of identity [41]. One study that was conducted during the COVID-19 pandemic demonstrated that teachers lack technological experiences and knowledge [42]. Furthermore, our results are consistent with findings from other studies demonstrating that inadequate training in teachers' digital skills may lead to negative emotions, and that these experiences differed between teachers, their age and school characteristics. Therefore, it is recommended that more professional development instructions should be available for PE teachers to enhance technological ability. This training should focus on technical skills as well as understanding how technology can be integrated into teaching lessons [24,25,42].

Moreover, teacher training must be seen as an essential factor to reverse such feelings and perspectives. In the coming years, teacher training should necessarily consider new social circumstances and be able to support teachers in providing an effective response to their students in the situations that require remote teaching. Since teachers were not trained for this specific situation, policies should be committed to education in general and to PE in particular, in order to strengthen the professional identity of educators and allow them to face the difficulties already inherent in their profession in a better way [25].

*Strengths, Limitations and Directions for Future Study*

This study had many strengths. For the first time in Israel, the study findings generated suggestions for the field of education, schools, and PE teachers. It began to shed light on the teachers' professional and emotional needs during remote teaching. The research findings showed that teachers should have a distinct, clear remote policy and remote-teaching training that contain a supportive program that empowers teachers' wellbeing, self-resilience, and professional efficacy.

Professional development programs that are customized to PE teachers may enhance remote-teaching skills and knowledge, as well as boost their professional efficacy, which may help teachers to teach more effectively. In addition, using accessible professional development opportunities to increase skills and professional efficacy may lower job stress and enhance satisfaction.

This study had a few limitations. First, this was a convenience sample conducted by a self-reported online questionnaire. Second, the study focused mainly on secular state schools and did not include sufficient teachers from different religious schools, mostly due to the lack of responses of PE teachers from Arab and Druze schools. Future studies should include more heterogenic populations.

## 5. Conclusions

Our study demonstrated teachers' self-resilience by showing the association between remote-teaching experience and subjective health perspective. Particularly, female teachers with greater remote-teaching experience and higher self-resilience reported higher levels of subjective health perception. Our findings highlight the need for a clear remote-teaching policy to empower teacher professional efficacy. Transparent procedures and guidelines, modified to students' ages and abilities and given by supportive administration are important for conducting effective PE lessons.

**Author Contributions:** Conceptualization, R.T., R.B.A., G.G. and G.J.; data curation, S.L. and N.M.; data analysis and interpretation, K.N., S.B. and G.G.; writing—original draft, R.B.A. and R.T.; writing—review and editing, G.G., K.N., N.M. and Y.H. All authors have read and agreed to the published version of the manuscript.

**Funding:** This research was funded by grants from the Chief Scientist of the Ministry of Education. Funding number: 483/21.

**Institutional Review Board Statement:** The study was conducted and approved by the Institutional Ethics Committee of Ariel University (AU-HEA-RT-20211007).

**Informed Consent Statement:** Informed consent was obtained from all subjects involved in the study.

**Conflicts of Interest:** The authors declare no conflict of interest.

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
