# Peer review of "Remote Teaching, Self-Resilience, Stress, Professional Efficacy, and Subjective Health among Israeli PE Teachers during the COVID-19 Pandemic"

_education, doi:10.3390/educsci12060405_

Round 1

Reviewer 1 Report

This is a very interesting article. Its subject is very topical, because it reflects one of the key aspects of making education resilient, the situation and experience of teachers in the pandemic caused by COVID19.
Moreover, the fact that the article and the study are located in Israel provides us with relevant data that can be compared with other countries. We must bear in mind that the management of the pandemic in this country had special characteristics, such as being the most advanced and the first to use vaccines. 
In Spain, similar and very interesting studies can be found for comparison and citation, such as those presented by Professor Portillo, Professor Bilbao Quitana and Professor Trujillo.
I also recommend including the concept of Remote Emergency Education (REE) analysed, among others, by Hodges, since this term has been the way in which the scientific community has defined the type of teaching carried out during the hardest times of the pandemic. In this way, the theoretical framework would be enriched.
I consider this work to be of sufficient methodological quality and with well-represented and very interesting results.
I congratulate the authors on their research.

Author Response

This is a very interesting article. Its subject is very topical, because it reflects one of the key aspects of making education resilient, the situation and experience of teachers in the pandemic caused by COVID19.
Moreover, the fact that the article and the study are located in Israel provides us with relevant data that can be compared with other countries. We must bear in mind that the management of the pandemic in this country had special characteristics, such as being the most advanced and the first to use vaccines. 
In Spain, similar and very interesting studies can be found for comparison and citation, such as those presented by Professor Portillo, Professor Bilbao Quitana and Professor Trujillo.
I also recommend including the concept of Remote Emergency Education (REE) analyzed, among others, by Hodges, since this term has been the way in which the scientific community has defined the type of teaching carried out during the hardest times of the pandemic. In this way, the theoretical framework would be enriched.
I consider this work to be of sufficient methodological quality and with well-represented and very interesting results.
I congratulate the authors on their research.

Thank you for your comment. We have added the following paragraph to better describe the reviewer's request in the introduction section [Lines 44-53].

Reviewer 2 Report

El trabajo es correcto, con un planteamiento limitado pero bien acotado en cuanto a definición de objetivos y consecuencias de los resultados.

(The work is correct, with a limited approach but well delimited in terms
of defining objectives and consequences of the results.)

Author Response

Review 2:

(The work is correct, with a limited approach but well delimited in terms
of defining objectives and consequences of the results.)

 Thank you for your comment, we appreciate it.

Reviewer 3 Report

Increase bibliographic references to other international researches carried out on the same topics.

I consider it appropriate to expand the bibliographic search, with references to international scientific articles. A lot has been written about this topic and it is important to mention it, adding some reflections in the article as well, which motivate the choices made in the research.

Author Response

Review 3

Increase bibliographic references to other international researches carried out on the same topics.

I consider it appropriate to expand the bibliographic search, with references to international scientific articles. A lot has been written about this topic and it is important to mention it, adding some reflections in the article as well, which motivate the choices made in the research.

Thank you for your comment. We have added the following paragraph to better describe the reviewer's request: in the introduction section, [ Lines 44-53] with references to international scientific articles.